# Dietary Potassium and Clinical Outcomes among Patients on Peritoneal Dialysis

**DOI:** 10.3390/nu15194271

**Published:** 2023-10-06

**Authors:** Jinru Pan, Xiao Xu, Zi Wang, Tiantian Ma, Jie Dong

**Affiliations:** 1Renal Division, Department of Medicine, Peking University First Hospital, Peking University Institute of Nephrology, Beijing 100871, China; jinru_pan@stu.pku.edu.cn (J.P.); xuxiaobj@bjmu.edu.cn (X.X.); tonypedia@126.com (Z.W.); matiantian1986@sina.com (T.M.); 2Key Laboratory of Renal Disease, Ministry of Health, Beijing 100034, China; 3Key Laboratory of Renal Disease, Ministry of Education, Beijing 100034, China

**Keywords:** chronic kidney disease, dietary potassium, mortality, peritoneal dialysis

## Abstract

Background: The association between dietary potassium and clinical prognosis is unclear in patients with chronic kidney disease (CKD). Here, we explored the association between dietary potassium intake and all-cause and cardiovascular (CV) mortality in peritoneal dialysis (PD) patients. Methods: Here, we present a retrospective analysis of a prospective study. Patients that began incident PD in our center between 1 October 2002 and 31 August 2014 were screened. We recorded all demographic and clinical data at baseline. Repeated measurements were recorded at regular intervals to calculate time-averaged values. Spline regression analysis and Cox proportional regression models were used to evaluate the relationship between dietary potassium and mortality. Results: We followed 881 PD patients for 45.0 (21.5, 80.0) months; 467 patients died, of which 189 (40.5%) died of CV death and 93 were still on PD treatment. Compared with those who had baseline dietary potassium ≥1200 mg/d, the majority of patients with lower dietary potassium were female, older, or poorly educated. They were prone to have poorer nutritional status, CV disease, and diabetes mellitus (*p* < 0.05). In the unadjusted analysis, both baseline and time-averaged dietary potassium <1200 mg/d predicted higher all-cause and CV mortality (*p* < 0.001~0.01). After adjusting for demographic and laboratory data, the association between potassium intake and all-cause and CV mortality weakened, which even disappeared after additional adjustment for dietary fiber, protein, and energy intake. Conclusions: Dietary potassium in PD patients was not independently associated with all-cause and CV mortality.

## 1. Introduction

Nutrition management is a key part of maintaining nutrition and metabolism stability among chronic kidney disease (CKD) patients, considering their complex mechanism for electrolyte disturbances and protein energy wasting [1,2]. Macronutrients and micronutrients intake, as the basis for nutrition maintenance, are mostly provided by natural foods for most CKD subjects with normal feeding ability and gastrointestinal function [1,2]. Among these nutrients, dietary potassium intake and its impacts on the clinical prognosis of CKD patients have received extensive attention. 

To date, the association between dietary potassium, serum potassium, and clinical prognosis are inconsistent in CKD patients [3,4,5,6,7,8,9,10]. Relevant studies with sample sizes ranging from 30 to 8000 have indicated that higher potassium intake was not or weakly correlated with hyperkalemia in non-dialysis CKD or HD patients [3,5,7,8,9,10], while a small cross-sectional study from Thailand showed that lower dietary potassium was a major contributor to hypokalemia in chronic peritoneal dialysis (PD) patients [4]. Studies have also indicated that higher dietary potassium was associated with a higher [10] or lower risk of death [6], or was not associated with mortality [8] in patients undergoing hemodialysis (HD). However, no study has explored the association between dietary potassium and death risk in non-dialysis CKD and PD patients. Of note, these studies enrolled participants with varied renal function and dietary patterns from different geographics and racial regions [3,4,5,6,8,10,11,12]. Dietary potassium was mostly assessed by food frequency questionnaires [3,6,8,10], a less reliable method compared with 3-day dietary record [13]. The observation period of study also differed when several endpoints were considered, i.e., 8 weeks to 5 years [3,4,5,6,8,10,11,12]. All these factors may explain the inconsistency of available evidences.

Among patients undergoing PD, the prevalence of hypokalemia ranges from 3% to 47% [14,15,16] and correlates with a high risk for peritonitis and higher mortality [14,15,16,17,18,19,20]. Hypokalemia might be due to common nutrients deficit in PD subjects [14,16,20], and no potassium is included in the dialysate. However, at present, whether hypokalemia is influenced by dietary potassium is unknown. On the other hand potassium-rich foods usually contain higher fiber and plant protein, which exert benefits on the gut microbiome and are also proven to be associated with decreased peritonitis and death risk in PD patients [21,22,23]. All these factors make it difficult to guide appropriate amounts of dietary potassium restriction among PD patients. 

Therefore, in this study, we prospectively explored the association between dietary potassium measured from 3-day dietary records and all-cause and cardiovascular (CV) mortality in PD patients. Additionally, among patients with different baseline dietary potassium values, we explored the trend of serum potassium during follow-up. This would help us understand the association of dietary potassium and clinical outcomes in the PD population. 

## 2. Methods

### 2.1. Study Patients

Here, we present a retrospective analysis of a prospective study, which was conducted at the PD center of Peking University First Hospital. We included incident PD patients in our center from 1 October 2002 to 31 August 2014. We excluded patients who were not end-stage kidney disease (ESKD), missed examination as requested, or could not be followed regularly. Each patient was followed until death, renal transplantation, transfer to HD, loss to follow-up, or the end of study (31 December 2020). Within 1 month after the implantation of PD catheter, all patients started the PD treatment and were provided lactate-buffered glucose dialysate with a twin-bag connection system (Baxter Healthcare, Guangzhou, China). Each patient received continuous ambulatory peritoneal dialysis treatment, and a physician visited them at least once every 3 months. This study was approved by the Medical Ethics Committee of Peking University (Project-ID: 2018[100], 27 June 2018).

### 2.2. Data Collection

Within one week prior to implantation of the PD catheter, we collected demographic data, such as age, gender, body mass index (BMI), and education status. Clinical data, including the presence of diabetes mellitus (DM) and CV disease, were also collected. Baseline values were averages of all measurements in the first 3 months, which included blood pressure, biochemistry tests, dialysis adequacy, and dietary and nutrition parameters. We calculated the averages of dietary nutrients in the first 6 months as baseline values. During follow-up, we prospectively collected all the above measurements and averaged them every 6 months to calculate time-averaged values.

### 2.3. Biochemical and Peritoneal Dialysis Parameters

We used an automatic Hitachi chemistry analyzer (Hitachi Chemical, Tokyo, Japan) to test biochemistry data, such as hemoglobin, serum albumin, creatinine, calcium, phosphate, lipids spectrum, and intact parathyroid hormone (iPTH). We measured serum high-sensitivity C-reactive protein (Hs-CRP) using immune rate nephelometric analysis. Via the collection of dialysate and 24 h urine, we measured dialysis adequacy, residual renal function (RRF), and glucose absorption. We defined dialysis adequacy as total urea nitrogen clearance (total KT/V) and total creatinine clearance (total CCr). The average renal clearance of creatinine and urea nitrogen were used to estimate RRF.

### 2.4. Dietary Assessment

During follow-up, patients finished 3-day dietary records before visiting the dietitian. Food models were used by a dedicated dietitian to check the diary. If the dietitian found that patients recorded diet diary in less than 3 days or the dietary records were not successfully checked, that would be invalid. The dietitian estimated actual amounts of foods in the diary records by use of food models. A computer software program (PD Information Management System, Peritoneal Dialysis Center, Peking University, Beijing, China) was used to calculate daily fiber, energy, protein, carbohydrate, fat, and potassium. To calculate the total amount of energy and protein intake, oral nutrition supplements were recorded as well. The daily energy intake includes energy obtained from dietary and dialysate. We measured glucose absorption as the difference between the amount of glucose in peritoneal dialysate absorbed into the peritoneal cavity in 24 h and that measured in the 24 h drained effluent, which was presented as grams of glucose/day (g/d). Similarly, we calculated dialysate energy absorption as kcal of energy/day (kcal/d).

### 2.5. Outcomes

The outcomes of interest were all-cause and CV death. We defined the CV death as death due to congestive heart failure, arrhythmia, myocardial infarction, cerebral infarction, cerebral bleeding, peripheral arterial disease, and sudden death. We censored follow-up at renal transplantation, transferring to HD, loss to follow-up, or the end of the study in all analyses (31 December 2020).

### 2.6. Statistical Analysis

We used SPSS 26.0 and R 4.2.2 for statistical analyses. We expressed continuous data as mean ± standard deviation. Categorical variables were presented as percentages or ratios and we expressed nonparametric data as median values with an inter-quartile range (IQR). One-way ANOVA, Kruskal–Wallis, or the X^2^ test were used to compare the differences of variables between groups, and then we adjusted the variables which differed significantly in Cox proportional hazard regression analysis. The cut-off point for grouping was determined via spline regression analysis showing the point associated with the increased all-cause and CV mortality.

Potential confounders combined with dietary potassium were examined by Cox proportional regression models to indicate the all-cause and CV death risk in the prospective analysis. Hazard ratios (HR) with 95% confidence intervals (CI) were calculated to present results of the Cox regression analysis. Analyses for baseline dietary potassium were adjusted for age, gender, BMI, educational level, DM, CV disease, and diastolic blood pressure (DBP) (Model 1), and further adjusted for albumin, serum creatinine, serum potassium, serum phosphorus, total cholesterol, peritoneal dialysis KT/V, residual kidney KT/V, and Hs-CRP (Model 2). Taking the possible influence of dietary nutrients into account, we further adjusted for fiber intake, protein intake, and energy intake (Model 3). Accordingly, analysis for time-averaged potassium intake were adjusted for time-averaged variables. 

Restricted cubic splines (RCS) fitted for multivariable-adjusted Cox regression were used to visualize the relationship between dietary potassium and all-cause and CV mortality. Three knots at the 15th, 50th, and 85th percentiles of dietary potassium were used in cubic splines. We used half-yearly measurements to calculate time-averaged biochemistry, dialysis, and nutrition parameters in the models. In addition, a 3-year observation period was chosen to calculate the time-averaged values. Significance level was set at 0.05, and statistical tests were two-sided.

## 3. Results

### 3.1. Baseline Characteristics and Follow-Up

Table 1 presented the baseline characteristics. We followed 881 PD patients with ESRD (434 men), mean age of 57.7 ± 14.8 years; 42.6% had CV disease, and DM was present in 42.1%. In the overall cohort, the mean ± SD of baseline daily dietary potassium was 1470.3 ± 435.3 mg/d.

The median follow-up duration was 45.0 (21.5, 80.0) months. At the end of the study, 467 patients died, of which 189 (40.5%) died of CV death, 177 had transferred to HD, 121 had received renal transplantation, and 93 were still on PD treatment (Figure 1). 

### 3.2. Dietary Potassium and Clinical Characteristics

According to spline regression analysis, we observed approximate L-shaped associations between dietary potassium and clinical prognosis first. Baseline or time-averaged dietary potassium <1200 mg/d was significantly associated with the increased all-cause and CV mortality, respectively (*p* < 0.001 for both), in the unadjusted model, as shown in Figure 2 and Figure 3. Therefore, we divided patients into two groups using the cut-off value of dietary potassium, i.e., 1200 mg/d (Table 1 and Appendix A). Compared with those who had baseline dietary potassium ≥1200 mg/d, the majority of patients with lower dietary potassium were female, older, and poorly educated, and they were prone to having poorer nutritional status. They also had lower DBP, but a higher prevalence of CV disease and DM (*p* < 0.05). The lower potassium intake was also associated with the decreased serum albumin, urea nitrogen, serum creatinine, serum potassium, and serum phosphorus (*p* < 0.05). Additionally, patients who had lower baseline dietary potassium had poorer RRF and serum lipid spectrum and severe inflammation status (*p* < 0.05). Notably, except for fatty acid, intakes of all nutrients, including energy intake, protein intake, fat intake, and carbohydrate intake, decreased along with the decreased baseline dietary potassium (*p* < 0.001). Patients with lower time-averaged dietary potassium also presented similar trends.

### 3.3. Predictive Value of Dietary Potassium for Mortality

As shown in Table 2, patients with baseline dietary potassium intake <1200 mg/d had a shorter follow-up time (*p* < 0.001) and higher mortality rate (*p* < 0.001), mostly due to cardiovascular events (*p* < 0.001), severe malnutrition (*p* < 0.001), and gastrointestinal hemorrhage (*p* = 0.012). There was no statistical difference in the rate of transferring to HD between the two groups. 

We analyzed the association between baseline or time-averaged dietary potassium and prognosis, respectively (Table 3). Compared with the group of dietary potassium ≥1200 mg/d, lower baseline and time-averaged dietary potassium were significantly associated with an 80.5% (*p* < 0.001) and 56.9% (*p* < 0.001) increase in all-cause mortality in the unadjusted analysis, respectively. This effect remained significant after adjusting for age, sex, BMI, education, DM, CV disease, and DBP, but weakened after additionally adjusting for biochemistry data such as serum albumin, creatinine, phosphorus, potassium, cholesterol, dialysis, and renal KT/V and Hs-CRP. After additionally adjusting for dietary nutrients including protein intake, energy intake, and fiber intake, the association between potassium intake and all-cause mortality disappeared. 

As for CV mortality, compared with the group of dietary potassium ≥1200 mg/d, lower baseline and time-averaged dietary potassium were significantly associated with a 102.3% (*p* < 0.001) and 47.3% (*p* = 0.01) increase in CV mortality in the unadjusted analysis, respectively. However, this effect weakened after adjustment for multiple models.

Visual models of dietary potassium and all-cause and CV mortality were established using RCS (Figure 2 and Figure 3). According to RCS curves, baseline and time-averaged dietary potassium were significantly associated with all-cause and CV mortality in patients undergoing PD without adjustment for any variables. With the increase in potassium intake, there was a significant reduction in death risk. However, after adjusting for Model 3, the HR curve showed an upward trend without significance, which verified again that fiber intake, energy intake, and protein intake were key confounders in the relationship between dietary potassium and mortality among PD patients.

### 3.4. Comparisons in the Prevalence of Hyperkalemia and Hypokalemia during the Follow-up between Groups by Baseline Dietary Potassium

As shown in Figure 4, the distribution of serum potassium between patients with baseline dietary potassium <1200 and ≥1200 mg/d was significantly different at each time point during follow-up, except at the 36th month (*p* < 0.05). Compared with those with potassium ≥1200 mg/d, patients with potassium <1200 mg/d were more likely to have hypokalemia, i.e., 6.3~13.6%, if defined as serum potassium <3.5 mmol/L, at each time point during follow-up. Despite that, patients with potassium ≥1200 mg/d were also prone to have hyperkalemia at the 6th, 18th, 24th, 30th, and 36th month, if defined as serum potassium >5.5 mmol/L, and the prevalence was relatively low, i.e., 2.6~6.3%. 

## 4. Discussion

In our long-term prospective cohort of PD patients, we found that baseline and time-averaged dietary potassium <1200 mg/d ascertained by 3-day dietary records were associated with a higher all-cause and CV mortality. However, after adjustment for nutrition indices, dialysis adequacy, inflammation status, and dietary nutrients, these associations weakened or disappeared. Our findings suggested that the relationship between dietary potassium and clinical outcomes is confounded by the above variables, especially nutritional indices and nutrients intake. 

Our PD subjects with low dietary potassium were older and had comorbidities with worse RRF and nutritional status. These clues have been observed repeatedly by previous studies [6,8,24]. It is also reasonable to find that decreased potassium intake was accompanied by other nutrient deficits in our data. After adjusting for dietary protein, energy, and fiber, the association between dietary potassium and prognosis disappeared. As supported by our previous studies, protein and energy intake were also important confounders for the relationship between dietary fiber or plant protein ratio and clinical outcomes in PD patients [22,23]. All these suggest that the prognostic value of any single dietary nutrient from regularly mixed foods is hard to be separated from other nutrients. Similarly, a recent study performed in HD patients from Europe and South America also indicated that a higher potassium intake is not independently associated with hyperkalemia or mortality after adjustment for total energy intake and food groups [8]. Therefore, our finding could be explained as dietary protein, energy, and fiber intake exerting a stronger impact on clinical outcomes, rather than neglecting the value of dietary potassium. 

In general population, the Dietary Approaches to Stop Hypertension (DASH) diet, rich in vegetables and fruits, showed that dietary potassium predicted a reduced risk of CV death and progression to ESKD [25,26]. In contrast, in the NIED study, dietary potassium mainly from chicken and beef was associated with an increased death risk in patients undergoing HD [10], which might due to the increased risk of renal hyperfiltration and the rapid decline in kidney function caused by higher animal protein intake [3,27]. These two findings again suggest the confounding effect of various macronutrients and micronutrients on potassium intake. In addition, potassium content from the same foods may have different preparation methods and potassium bioavailability [3]. All these lead to the phenomenon that dietary potassium under different dietary patterns may have varied impacts on the clinical outcomes of patients. 

In our prospective study, we observed that PD patients with baseline dietary potassium <1200 mg/d had a higher incidence of hypokalemia during the follow-up period, i.e., 6.3~13.6%, which was numerically higher than the incidence of hyperkalemia among those with baseline dietary potassium ≥1200 mg/d, i.e., 2.6~6.3%. In the PD population, several studies have consistently reported the negative impact of hypokalemia on arrhythmia-related and all-cause mortality, infectious-caused mortality, or subsequent peritonitis risk [14,15,18,19]. The population-attributable risks for all-cause mortality for serum potassium <4.0 were 3.6%, numerically higher than 1.9% for serum potassium ≥5.5 mEq/L, respectively [15]. Therefore, we suggest sufficient potassium intake from natural foods be ensured, combined with oral potassium supplements if needed, in order to achieve a safe range of serum potassium in clinical practice. Considering potassium loss and potassium distribution across extracellular and intracellular fluid markedly varies, we support the recommendation from the KDIGO guideline, i.e., a more individualized approach should be taken to dietary potassium in CKD patients instead of providing a uniform recommendation [13]. A recent randomized controlled trial, performed in PD patients who have hypokalemia, showed that maintaining a serum potassium concentration within 4–5 mEq/L with protocol-based oral potassium supplement may reduce the risk of peritonitis [28]. Further studies are needed to verify the benefits of potassium supplementation in correcting hypokalemia and if this would in turn improve the final outcomes. 

Our study has several advantages. To our knowledge, we are the first to examine the relationship between dietary potassium and clinical prognosis in PD patients. We repeatedly collected complete data on diet, nutrition, and biochemistry at multiple time points, which gave us an opportunity to better test our hypothesis. In addition, long-term follow-up and large quantities of outcome events provided adequate statistical power. Compared with previous studies, we used 3-day dietary records to evaluate nutrients intake, which is considered to be more precise than food frequency questionnaires in stage 3–5 CKD during both dialysis and nondialysis treatment days [13]. 

Some limitations should be considered in interpreting our findings. As with all observational studies, our study cannot determine a causal association between dietary potassium and clinical prognosis in PD patients. In addition, potassium additives and oral potassium supplements were not evaluated in our study, which could underestimate the amount of potassium intake, concealing its potential link with mortality. Last, we conducted a single-center study, so our findings and our conclusion cannot be extended to other populations and regions.

In conclusion, through our prospective and single-center PD cohort, dietary potassium was not independently associated with all-cause and CV mortality. Since significantly confounding effects of nutrition indices and mixed nutrients on dietary potassium exist, the inconsistent and paradoxical relationship between dietary potassium and clinical outcomes is explainable. Since current evidences could not help us reach a consensus on the exact amount and appropriate source of dietary potassium, individualized potassium intake for maintaining a safe range of serum potassium for the prevention of hypokalemia and hyperkalemia could be advisable in the setting of PD.

## Figures and Tables

**Figure 1 nutrients-15-04271-f001:**
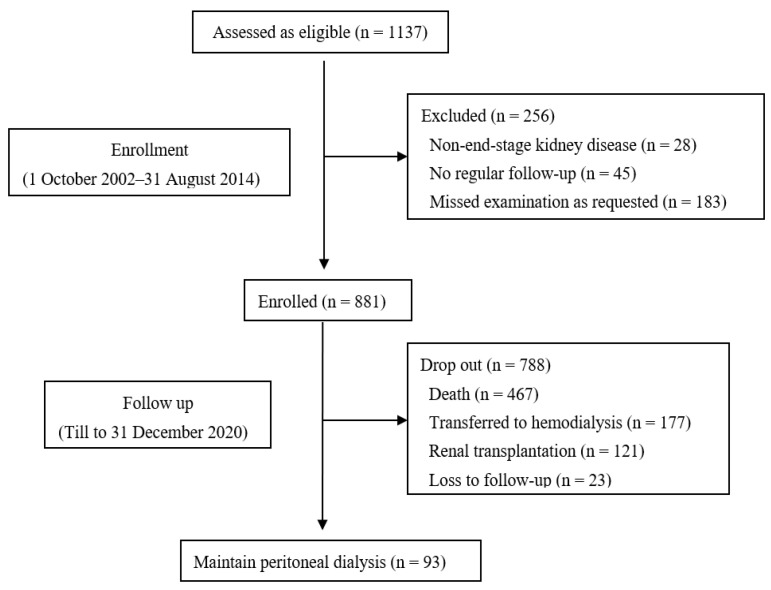
Flow chart.

**Figure 2 nutrients-15-04271-f002:**
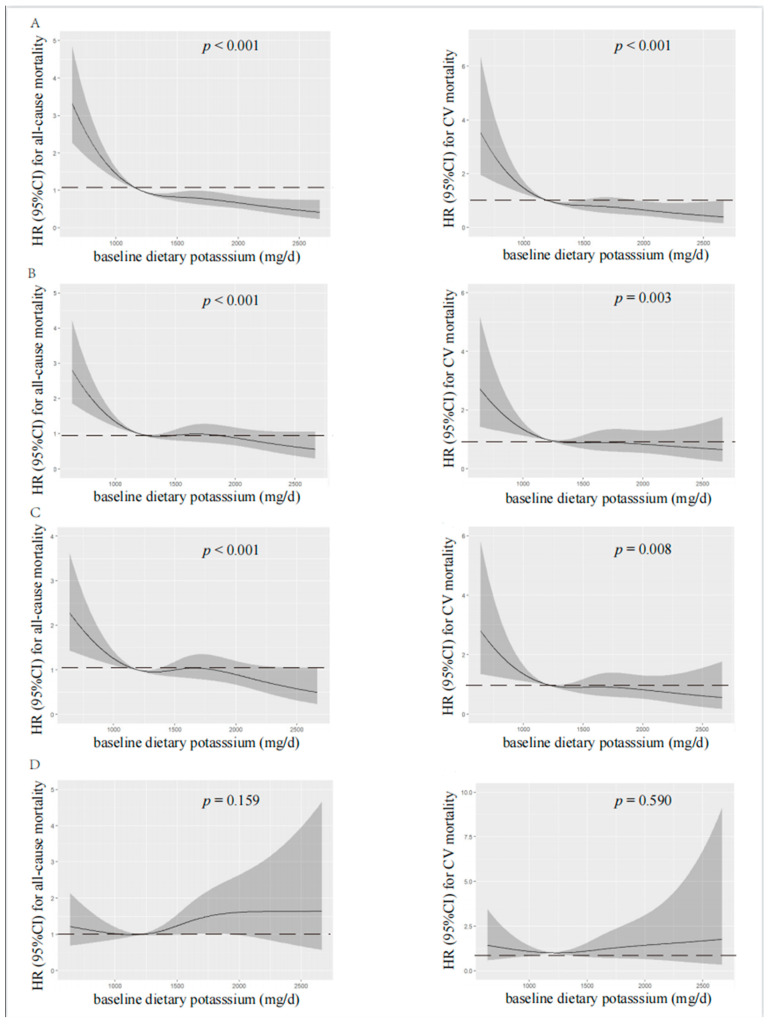
Restricted cubic spline analysis with multivariate-adjusted associations between baseline dietary potassium and all-cause (**left**) and CV (**right**) mortality. (**A**) Unadjusted. (**B**) Adjusted for Model 1 including age, gender, BMI, educational level, DM, CV disease, and DBP. (**C**) Adjusted for Model 2 including Model 1 plus albumin, serum creatinine, serum potassium, serum phosphorus, total cholesterol, peritoneal dialysis KT/V, residual kidney KT/V, and Hs-CRP. (**D**) Adjusted for Model 3 including Model 2 plus fiber, protein, and energy intake. HR, hazard ratio; CI, confidence interval; CV, cardiovascular.

**Figure 3 nutrients-15-04271-f003:**
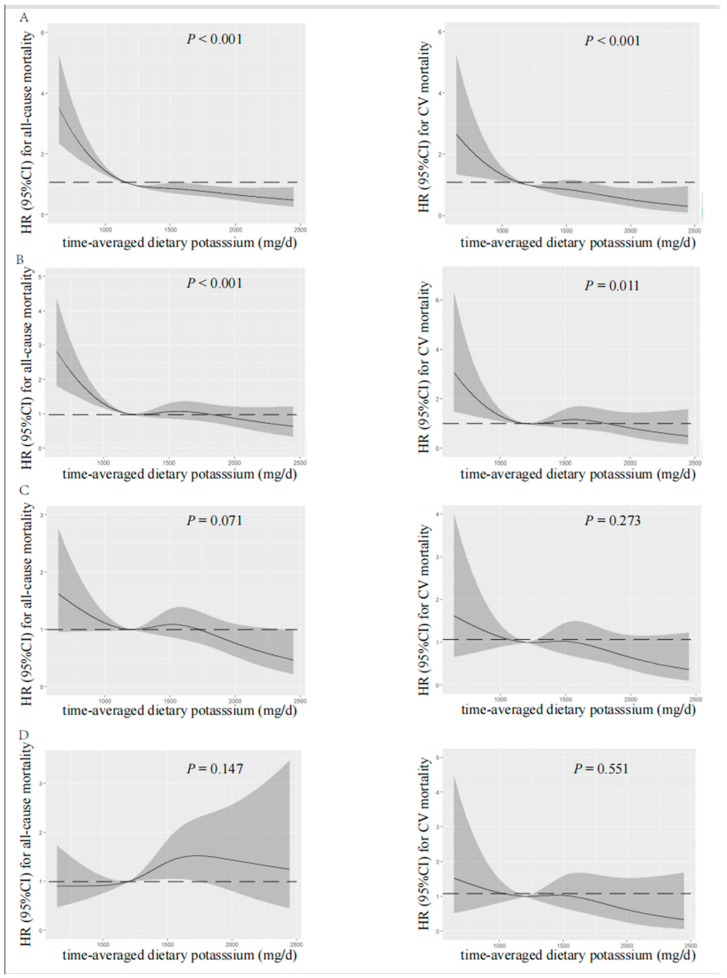
Restricted cubic spline analysis with multivariate-adjusted associations between time-averaged dietary potassium and all-cause (**left**) and CV (**right**) mortality. (**A**) Unadjusted. (**B**) Adjusted for Model 1 including age, gender, time-averaged BMI, educational level, DM, CV disease, and time-averaged DBP. (**C**) Adjusted for Model 2 including Model 1 plus time-averaged albumin, time-averaged serum creatinine, time-averaged serum potassium, time-averaged serum phosphorus, time-averaged total cholesterol, time-averaged peritoneal dialysis KT/V, time-averaged residual kidney KT/V, and Hs-CRP. (**D**) Adjusted for Model 3 including Model 2 plus time-averaged fiber intake, time-averaged protein intake, and time-averaged energy intake. HR, hazard ratio; CI, confidence interval; CV, cardiovascular.

**Figure 4 nutrients-15-04271-f004:**
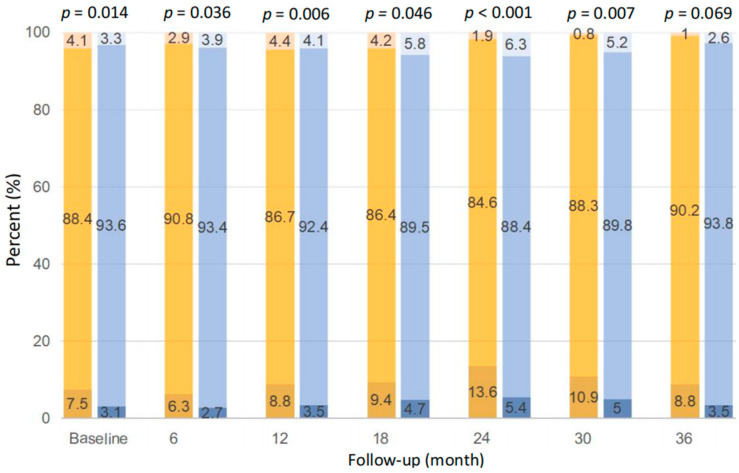
Comparison of serum potassium distribution between dietary potassium <1200 mg/d (left bar at each time point) and ≥1200 mg/d (right bar at each time point) at different time points. The three colors of the bars from bottom to top indicate serum potassium <3.5, 3.5–5.5, and >5.5 mmol/L, respectively.

**Table 1 nutrients-15-04271-t001:** Baseline clinical characteristics of peritoneal dialysis patients according to dietary potassium at baseline (*n* = 881).

Characteristic	Total	Baseline Dietary Potassium		*p*
		<1200 mg/d	≥1200 mg/d	
Age, years	57.7 ± 14.8 ^1^	61.9 ± 14.6	56.1 ± 14.6	<0.001
Male, n (%)	434 (49.3)	89 (36.8)	345 (54.0)	<0.001
BMI ^2^, kg/m^2^	23.3 ± 3.7	23.2 ± 4.0	23.3 ± 98.9	0.660
Educational level				<0.001
≤Elementary school, n (%)	149 (16.9)	61 (25.2)	88 (13.8)	
Middle school, n (%)	229 (26.0)	69 (28.5)	160 (25.0)	
High school, n (%)	244 (27.7)	57 (23.6)	187 (29.3)	
>High school, n (%)	259 (29.4)	55 (22.7)	204 (31.9)	
DM, n (%)	371 (42.1)	118 (48.8)	253 (39.6)	0.014
CV disease, n (%)	375 (42.6)	131 (54.1)	244 (38.2)	<0.001
SBP, mmHg	135.8 ± 16.7	136.8 ± 18.4	135.4 ± 16.0	0.274
DBP, mmHg	79.0 ± 11.2	77.4 ± 11.0	80.0 ± 11.2	0.010
MAP, mmHg	98.0 ± 11.2	97.4 ± 11.4	98.3 ± 11.1	0.271
Laboratory and nutritional data				
Albumin, g/L	35.4 ± 4.6	34.4 ± 4.8	35.7 ± 4.5	<0.001
Hemoglobin, g/L	102.8 ± 15.7	102.0 ± 15.2	103.1 ± 15.8	0.344
Hs-CRP, mg/L	2.1 (0.7, 5.7)	2.9 (1.1, 8.4)	1.9 (0.7, 5.0)	<0.001
Urea nitrogen, mmol/L	22.5 ± 6.1	20.7 ± 6.0	23.2 ± 6.1	<0.001
Serum creatinine, μmol/L	689.2 ± 233.6	655.0 ± 224.0	702.1 ± 236.1	0.008
Serum calcium, mmol/L	2.2 ± 0.2	2.2 ± 0.2	2.2 ± 0.2	0.154
Serum phosphorus, mmol/L	1.6 ± 0.4	1.5 ± 0.4	1.6 ± 0.4	<0.001
Serum potassium, mmol/L	4.4 ± 0.6	4.3 ± 0.6	4.5 ± 0.6	<0.001
Serum sodium, mmol/L	139.2 ± 3.0	139.2 ± 2.8	139.2 ± 3.1	0.787
HDL-cholesterol, mmol/L	1.1 ± 0.3	1.2 ± 0.4	1.1 ± 0.3	0.373
LDL-cholesterol, mmol/L	2.6 ± 0.8	2.6 ± 0.8	2.6 ± 0.8	0.586
Total cholesterol, mmol/L	4.9 ± 1.1	5.0 ± 1.1	4.8 ± 1.2	0.018
Triglycerides, mmol/L	1.5 (1.1, 2.0)	1.6 (1.2, 2.2)	1.5 (1.1, 2.0)	0.014
iPTH, pg/mL	164.1 (77.4, 320.7)	159.4 (68.4, 290.5)	169 (80.0, 324.6)	0.185
Total CCr, L/w/1.73 m^2^	72.8 ± 27.9	69.7 ± 25.0	74.0 ± 28.9	0.049
Total Kt/V	1.9 ± 0.5	1.9 ± 0.5	1.9 ± 0.6	0.650
RRF, mL/min	3.7 (2.1, 5.6)	3.3 (1.7, 5.4)	3.8 (2.3, 5.7)	0.011
Energy intake, kcal/day	1660.9 ± 334.3	1387.0 ± 246.0	1763.7 ± 303.9	<0.001
Protein intake, g/day	52.1 ± 13.9	39.7 ± 8.7	56.9 ± 12.5	<0.001
Fat intake, g/day	54.0 ± 14.5	44.7 ± 12.8	57.6 ± 13.5	<0.001
Carbohydrate intake, g/day	184.6 ± 49.2	146.9 ± 35.0	198.9 ± 46.1	<0.001
Fiber intake, g/day	8.2 ± 3.4	5.4 ± 2.1	9.2 ± 3.2	<0.001
Fatty acid intake, g/day	43.0 ± 12.2	42.9 ± 12.0	43.3 ± 12.3	0.726
Potassium intake, mg/day	1470.3 ± 435.3	980.6 ± 177.0	1655.7 ± 352.5	<0.001
nDEI, kcal/kg/d	28.5 ± 5.4	25.0 ± 4.9	29.8 ± 5.0	<0.001
nDPI, g/kg/d	0.85 ± 0.24	0.7 ± 0.2	0.9 ± 0.2	<0.001

^1^ Values are expressed as mean ± standard deviation, percentage or median with upper and lower quartile or percentage. ^2^ Abbreviation: BMI, body mass index; CV, cardiovascular; CCr, creatinine clearance; DM, diabetes mellitus; DBP, diastolic blood pressure; Hs-CRP, high-sensitivity C-reactive protein; HDL, high-density lipoprotein; iPTH, intact parathyroid hormone; Kt/V, urea clearance; LDL, low-density lipoprotein; MAP, mean arterial pressure; nDEI, normalized energy intake; nDPI, normalized protein intake; RRF, residual renal function; SBP, systolic blood pressure.

**Table 2 nutrients-15-04271-t002:** Outcomes among peritoneal dialysis patients based on dietary potassium at baseline (n = 881).

Outcomes, No. of Events (Event Rate/100 Person-Years)	Total	Dietary Potassium		*p*
		<1200 mg/d	≥1200 mg/d	
Follow-up, months	45.0 (21.5, 80.0)	33.0 (18.8, 62.3)	49.0 (23.0, 85.0)	<0.001
Maintain peritoneal dialysis	93 (2.30)	17 (1.84)	76 (2.43)	0.591
Death	467 (11.54)	159 (17.18)	308 (9.87)	<0.001
Cardiovascular events *	189 (4.67)	69 (7.46)	120 (3.84)	<0.001
Infection	115 (2.84)	30 (3.24)	85 (2.72)	0.252
Severe malnutrition	20 (0.50)	12 (1.30)	8 (0.26)	<0.001
Gastrointestinal hemorrhage	22 (0.54)	10 (1.08)	12 (0.38)	0.012
Tumor	44 (1.09)	15 (1.62)	29 (0.93)	0.074
Others	77 (1.90)	23 (2.49)	54 (1.73)	0.154
Transfer to hemodialysis	177 (4.37)	41 (4.43)	136 (4.36)	0.674
PD-related infection	105 (2.59)	22 (2.38)	83 (2.66)	0.885
Fluid overload	17 (0.42)	4 (0.43)	13 (0.42)	0.768
Inadequate solute clearance	13 (0.32)	3 (0.32)	10 (0.32)	0.911
Catheter dysfunction	2 (0.05)	1 (0.11)	1 (0.03)	0.242
Socioeconomic causes	22 (0.54)	9 (0.97)	13 (0.42)	0.049
Renal transplantation	121 (2.99)	20 (2.16)	101 (3.24)	0.042

* Cardiovascular events include cardiovascular events, cerebrovascular events, and sudden death.

**Table 3 nutrients-15-04271-t003:** Associations of baseline and time-averaged dietary potassium with all-cause and cardiovascular mortality (*n* = 881).

		Unadjusted		Model 1		Model 2		Model 3	
		HR (95%CI)	*p*	HR (95%CI)	*p*	HR (95%CI)	*p*	HR (95%CI)	*p*
All-cause mortality									
Dietary potassium (mg/d)									
Baseline	≥1200	reference		reference		reference		reference	
	<1200	1.805 (1.490, 2.188)	0.001	1.313 (1.065, 1.618)	0.011	1.226 (0.982, 1.531)	0.072	0.880 (0.663, 1.166)	0.373
Time-averaged	≥1200	reference		reference		reference		reference	
	<1200	1.569 (1.302, 1.890)	<0.001	1.311 (1.072, 1.603)	0.008	1.121 (0.899, 1.397)	0.310	0.844 (0.635, 1.121)	0.242
Cardiovascular mortality									
Dietary potassium (mg/d)									
Baseline	≥1200	reference		reference		reference		reference	
	<1200	2.023 (1.502, 2.275)	<0.001	1.393 (1.009, 1.923)	0.044	1.407 (1.000, 1.980)	0.050	0.987 (0.639, 1.523)	0.952
Time-averaged	≥1200	reference		reference		reference		reference	
	<1200	1.473 (1.097, 1.979)	0.010	1.241 (0.904, 1.705)	0.182	1.155 (0.817, 1.631)	0.414	0.992 (0.645, 1.526)	0.972

Model 1: Adjusted for age, gender, BMI, educational level, DM, CV disease, and DBP. Model 2: Adjusted for Model 1 plus albumin, serum creatinine, serum potassium, serum phosphorus, total cholesterol, peritoneal dialysis KT/V, residual kidney KT/V, and Hs-CRP. Model 3: Adjusted for Model 2 plus fiber, protein, and energy intake. Analysis for time-averaged dietary potassium was adjusted for time-averaged variables accordingly. HR, hazard ratio; CI, confidence interval.

## Data Availability

Due to the Management of China’s Human Genetic Resources, the data described in the manuscript, code book, and analytic code will not be available.

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
