# Peer review of "Dietary Potassium and Clinical Outcomes among Patients on Peritoneal Dialysis"

_nutrients, 2023, doi:10.3390/nu15194271_

Round 1

Reviewer 1 Report

This is a retrospective analysis of a prospective study. The AA must give the reference of the previous published papers related to the original study 

the design prevents from drawing solid conclusions, so the consistency of the results are questionable and the priority for publication is quite  low.  Neverthelss, additional information about potassium intake in peritoneal patients are given. 

the paper reads quite well

Author Response

Response to Reviewer 1 Comments

1. Summary

Thank you very much for taking the time to review this manuscript. Please find the detailed responses below and the corresponding revisions in the re-submitted files.

2. Questions for General Evaluation

Reviewer’s Evaluation

Response and Revisions

Does the introduction provide sufficient background and include all relevant references?

Yes

Are all the cited references relevant to the research?

Can be improved

Is the research design appropriate?

Must be improved

Are the methods adequately described?

Must be improved

Are the results clearly presented?

Yes

Are the conclusions supported by the results?

Can be improved

3. Point-by-point response to Comments and Suggestions for Authors

Comments 1: This is a retrospective analysis of a prospective study. The AA must give the reference of the previous published papers related to the original study. The design prevents from drawing solid conclusions, so the consistency of the results are questionable and the priority for publication is quite low. Nevertheless, additional information about potassium intake in peritoneal patients are given.

Response 1: Thank you for pointing this out. We have cited previously published articles in the discussion, with reference numbers 22 and 23, respectively. However, the study design has been fixed at the beginning of the study, and we described the study details in the methods part as much as possible. We will learn from the experience and make it as well as possible in the future study design.

4. Response to Comments on the Quality of English Language

Point 1: the paper reads quite well.

Response 1: Thank you for your comments.

5. Additional clarifications

No.

Reviewer 2 Report

This is a very extensive prospective cohort study by Pan et al. evaluating relationship between dietary potassium and clinical outcomes in chronic kidney disease patients. The article presents sufficient data, which is in line with the readers’ interest of Nutrients. The paper is well conceived and structured, but there are some minor comments before publication. Please take into account the following comments.

1.       Write the corresponding author instead correspondence: author:

2.       Abstract is too long. It should be shortened to a maximum of 200 words.

3.       Write p in italics and in lower case.

4.       Expand the introduction a bit - too brief and insufficiently explained why that particular study.

5.       2.1. ethical approval number is missing.

6.       Revise references according to the journal's instructions - the name of the journal must be in italics, the year of publication must be in bold.

Minor editing of English is necessary.

Author Response

Response to Reviewer 2 Comments

1. Summary

Thank you very much for taking the time to review this manuscript. Please find the detailed responses below and the corresponding revisions in the re-submitted files.

2. Questions for General Evaluation

Reviewer’s Evaluation

Response and Revisions

Does the introduction provide sufficient background and include all relevant references?

Must be improved

Are all the cited references relevant to the research?

Yes

Is the research design appropriate?

Yes

Are the methods adequately described?

Yes

Are the results clearly presented?

Yes

Are the conclusions supported by the results?

Yes

3. Point-by-point response to Comments and Suggestions for Authors

Comments 1: Write the corresponding author instead correspondence: author:

Response 1: Thank you for pointing this out. We agree with this comment and have revised it accordingly.

Comments 2: Abstract is too long. It should be shortened to a maximum of 200 words.

Response 2: Thank you for your comments, but we have streamlined the abstract section and further simplification may not accurately describe the study design.

Comments 3: Write p in italics and in lower case.

Response 3: Thank you for your comments, we have revised all the p values accordingly in the text, tables and figures.

Comments 4: Expand the introduction a bit - too brief and insufficiently explained why that particular study.

Response 4: Thank you for pointing this out. For details, see paragraph 2, line 50-61, page 2 and  paragraph 3, line 69-81, page 2 of the manuscript.

Comments 5: 2.1. ethical approval number is missing.

Response 5: Thank you for pointing this out. We have added the ethical approval number at the end of 2.1.

Comments 6: Revise references according to the journal's instructions - the name of the journal must be in italics, the year of publication must be in bold.

Response 6: Thank you for pointing this out. We have revised the forms of references according to the journal's instructions.

4. Response to Comments on the Quality of English Language

Point 1: Minor editing of English is necessary.

Response 1: We made some edits to the English language, as highlighted in the article.

5. Additional clarifications

No.

Reviewer 3 Report

The article is devoted to a topical topic, but the introduction is poorly written. There are no significant materials. It is necessary to rewrite the introduction and conclusion.

The article is devoted to a topical topic, but the introduction is poorly written. There are no significant materials. It is necessary to rewrite the introduction and conclusion.

Author Response

Response to Reviewer 3 Comments

1. Summary

Thank you very much for taking the time to review this manuscript. Please find the detailed responses below and the corresponding revisions in the re-submitted files.

2. Questions for General Evaluation

Reviewer’s Evaluation

Response and Revisions

Does the introduction provide sufficient background and include all relevant references?

Can be improved

Are all the cited references relevant to the research?

Can be improved

Is the research design appropriate?

Can be improved

Are the methods adequately described?

Yes

Are the results clearly presented?

Yes

Are the conclusions supported by the results?

Yes

3. Point-by-point response to Comments and Suggestions for Authors

Comments 1: The article is devoted to a topical topic, but the introduction is poorly written. There are no significant materials. It is necessary to rewrite the introduction and conclusion.

Response 1:Thank you for pointing this out. For details, see paragraph 2, line 50-61, page 2 and  paragraph 3, line 69-81, page 2 of the manuscript.

4. Response to Comments on the Quality of English Language

Point 1: The article is devoted to a topical topic, but the introduction is poorly written. There are no significant materials. It is necessary to rewrite the introduction and conclusion.

Response 1: Please refer to the response in Section 3 for details

5. Additional clarifications

No.
